# Pursuing Consultant Performance: The Roles of Sustainable Leadership Styles, Sustainable Human Resource Management Practices, and Consultant Job Satisfaction

**Afriyadi Cahyadi [1,2,\*], József Poór [1] and Katalin Szabó [1]**

1   Doctoral School of Economics and Regional Sciences, Hungarian University of Agriculture and Life Sciences, H-2100 Gödöllő, Hungary; poor.jozsef@uni-mate.hu (J.P.); szabo.katalin@uni-mate.hu (K.S.)
2   Faculty of Economics, Sriwijaya University, Inderalaya 30662, Indonesia
\*   Correspondence: cahyadi.afriyadi@phd.uni-mate.hu or afriyadicahyadi@fe.unsri.ac.id

**Abstract:** Human resource management (HRM) consultants have a paramount role in solving current company problems, especially amid the COVID-19 pandemic. They are professionals who work for HRM consulting firms. This research aims to investigate the positive effects of sustainable leadership styles and HRM practices on consultant job satisfaction and performance in firms in Indonesia. We standardized and tested seven hypotheses and engaged the Snowball sampling method for collecting primary data. Then, we sent a self-report questionnaire with 27 items to respondents. Four hundred consultants are the target respondents in cross-sectional data collection from the beginning of January to the beginning of February 2022. The confirmatory factor analysis has produced valid and reliable items in total. The goodness of fit test has issued a fit model. Hence, first, the chief executive officers' (CEOs) sustainable leadership styles and HRM practices positively affect consultant job satisfaction. Second, consultant job satisfaction positively affects consultant performance. Therefore, sustainable leadership styles and sustainable HRM practices are independent variables. Job satisfaction is the mediating one.

**Keywords:** sustainable leadership styles; sustainable human resource management practices; employee job satisfaction; employee performance; management consulting firms

## 1. Introduction

Sustainability for business is an increasingly popular topic among academics and practitioners that disclose economics to the environment and society [1]. The company must cooperate with internal and external stakeholders to interpret justice, humanity, and environmental friendliness in business [2]. It makes corporate social responsibility (CSR) obsolete [3]. Recent studies on business leadership and human resource management practices (HRMPs) have enfolded this topic. A demand that is now emerging is the leadership styles and HRMPs that support the sustainability of the company. Research on this probe is meager in bibliographic databases, such as Scopus, Web of Science, Google Scholar, or journal websites on leadership, HRM, and sustainability.

The scope of sustainable leadership leads to agreement on a long-term vision, corporate and community relations, ethical behavior, social responsibility, innovation, change, and stakeholders [4]. This leadership frames collaboration among stakeholders and promotes long-term advantages for the company [5]. Also, it conceives merit for those individuals, organizations, society, and ecology with an attraction to the present and future generations [6]. They exist because of every innovation and technology and the latest sustainable business process for a long-term company process [7]. Indeed, sustainable leadership styles (SLSs) can develop based on this leadership.

Sustainable HRMPs (SHRMPs) in the company can be from the prescribed HRM approach. HRM studies in companies have currently debated strategy [8,9]. Researchers

have expounded that SHRM in the company is an HR strategy for financial, social, ecological, work relational, and developmental arrangements [10]. It unifies and reinforces objectives, inputs, processes, and outputs [11], and concerns for employees, the environment, profitability, society, regulations, and fairness [12]. The company accommodates it into an image that designs a positive uniqueness in business contests and enhances employee self-concept [13]. HRMPs of the company apply significant CSR content [14]. Their motive promotes HRM strategies to foster a culture of trust, collaboration, and employee engagement [15].

Recent studies have revealed how leadership styles and HRMPs improve employee performance in companies. Furthermore, job satisfaction is a consequence of leadership styles and HRMPs. So, job satisfaction is a mediator in the relationship between SLSs and employee performance and between HRMPs and employee performance. Then, we collect literature about these theories and engage a quantitative study of the HRM consulting firm within the extent of HRM in Indonesia for this case.

Consulting companies have existed for more than a century or since the second industrial revolution in the United States [16]. Management consultation is advising and assisting with expertise in management issues [17], facilitated by management experts individually or within the company. The development of this industry is from the scientific management era: 1890, human relations: 1930, leadership/management: 1950, IT/ICT orientation: 1980, and emerging tendencies: 1990. This service is based on the competencies of the consultants to ameliorate the effects of business strategies, operational processes, and business performance [18]. The management consultants act as external advisors of their clients to help benefit from a particular goal, specified issues, or even unrecognized issues. They must scan the culture and structure of their clients [19]. The management consultant works under direct strain from their clients and the consulting firms that oversees them [20]. Full-time employment with a single employer was normative for decades. The transition from an industry-based economy to a knowledge-based economy has driven people to seek highly skilled concentrations, individualized services, and self-employment [21].

Amid the disruption of the COVID-19 pandemic, the management consulting firms recorded high growth by assisting their clients in reorganizing their business operations and digitizing their infrastructure. All industries now hire consultants for business growth, performance, and efficiency. SLSs and HRMPs are a curiosity when sustainability is a demand for the business competition. Leaders of the management consulting firms manage reciprocal relationships with their stakeholders [22]. In Indonesia, HRM consulting firms continued their previous activities during the COVID-19 pandemic and even expanded it with the help of growing digitization. Their websites massively promote their services, such as training, mentoring, research, and assessment. Mihalicz (2020) has described fifty countries with their Institutes of Management Consulting as members of the International Council of Management Consulting Institutes (IMCCI). However, Indonesia is not a part of it, so the research is not general for cases in other countries. ICMCI, the leading global body of management consultants, has defined a set of parameters that describes the environment for the management consulting industry and publishes these parameters in the National Consulting Index (NCI). Several other Asian countries with NCIs are Bangladesh, China, Taiwan, Hong Kong, India, Japan, South Korea, Philippines, Singapore, and Thailand [23].

The single leadership style is impractical to be executed in companies constantly. Successful leaders often combine their leadership and management style [24]. This research aims to attest the positive effect of the CEO SLSs and sustainable HRMPs in HRM consulting firms on the performance of the consultants who work in them with the mediating role of the consultant job satisfaction (CJS). First of all, this article relates sustainability, leadership styles, HRMPs, CJS, and consultant performance (CP). After that, it proves a model based on primary data collected through an online survey using a questionnaire from consultants at HRM consulting firms in Indonesia. The results present and discuss the statistical test. Finally, the paper presents conclusions about the causal relationships in the research model.

Theoretical and practical implications are also the final part, which is explained based on the results and discussion.

## 2. Literature Review

### 2.1. The HRM Consulting Firms

The HRM consulting firms have a positive social and economic impact that drives sustainable growth. HRM consultants focus on adding value to HRM practices in their client companies. Such advice is provided by traditional HRM consulting firms. However, nowadays, more and more general or boutique-like consulting companies are also offering beside other services (e.g., strategy, process, IT, etc.) in addition Human Resource professional consulting services [25]. They are practitioners, sources of knowledge, and leaders in thought and conceptualization for their clients. They see the managerial problem of clients as a challenge and an opportunity to improve the quality of the client's human resources related activities and processes. On the one hand, companies can employ HRM consultants as part of their internal organization and work entirely in the company. On the other hand, externals work for HRM consulting firms that provide support to companies in the areas of HRM, such as resourcing, training and development, compensation, occupational safety and health, and employee relations.

The consulting process starts from the initial stage, including the pre-project, the project, and the post-project steps [26]. The initial stage is when the client companies' management team realizes a problem within the companies and requires resolution by external consultants. All consultancy work is concerned with the emergence, existence, and recognition of a problem. In the pre-project phase, the problem to be investigated is defined. The project stage is when the consultants diagnose the company's problems, collect data, and form a comprehensive picture of the organization's vital activities to analyze the data. Finally, the consultant and the clients compare the actual results to the planned ones in the post project. The consultant delivers a final report and performs a final presentation [27].

The success of the HRM consulting industry in a country is influenced by many factors [23]. Client companies hire management consulting services to adopt their intelligence in dealing with changes, superior methods, and creativity. Scholars have agreed that HRM consulting is provided by trained and qualified persons, objectively and independently [28]. In Indonesia, the demand for HRM consulting services continues to increase when viewed from the growth of HRM consulting firms that offer their services. Market growth in this industry is increasing because modern digital infrastructure is continuously developing in Indonesia. Many HRM consulting firms have company blogs [29]. They market their blogs and websites to social media platforms, such as Facebook and Instagram. Even amid the COVID-19 pandemic, HRM consulting firms continue to thrive by providing online services in Indonesia with various methods that are adaptive to the COVID-19 pandemic. These firms in Indonesia have offered a remote work-oriented approach to HRMPs in client companies. They mushroomed even amid the COVID-19 pandemic. The majority of them promote their company services on Google and social media platforms.

### 2.2. Leadership Styles, HRMPs, CP, Sustainability

Leadership styles and HRMPs have interacted in determining employee motivation and productivity [30]. All leadership styles referred to are transactional, ethical, authentic, transformational, servant, shared, LMX, and inclusive, in which they interact with HRMPs with the characteristics of traditional, productivity-based, compliance-based, calculative, collaborative, high-involvement, and commitment based. Leadership has interacted with HR behavior and practices in increasing employee task performance [31]. Empowering leadership and HRM systems have interacted for knowledge-intensive teamwork [32]. In the context of task performance, responsible leadership has interacted with HRM [33]. The interaction between authentic leadership and HRMPs are typical, consensus, consistent, intended, actual, and perceived [34].

Leadership styles and HRMPs can significantly defend the company's sustainability. Researchers have revealed that ethical leadership is related to CSR [35,36]. Transformational leadership relates to CSR [37]. Inclusive leadership relates to CSR [38], and leadership is a controlling responsibility to CSR [33]. Thus, ethical leadership, transformational leadership, inclusive leadership, and responsible leadership can reflect sustainable leadership styles in companies. Three HRM approaches that can reflect SHRMPs in companies are collaborative HRM, high-involvement HRM, and commitment-based HRM. Scholars have agreed that collaborative HRM is a company and society link [39]. Individual commitment to the environment must begin with employee selection based on environmental criteria, and several studies agree that employee involvement in environmental management supports the company's environmental performance [40]. Thus, collaboration, involvement, and employee commitment are three aspects of HRMPs that reflect sustainability in the company. On the one hand, sustainability requires company leaders to apply ethics, transformation, inclusivity, and responsibility to support synergies between the economy, environment, and society. On the other hand, sustainability requires HR managers to apply collaboration, high involvement, and commitment.

### 2.2.1. Ethical Leadership, Transformational Leadership, Inclusive Leadership, and Responsible Leadership, CP

Business ethics supports the success of company sustainability in the economic, social, and environmental aspects [41]. Scholars named ethical leadership, which is ethical behavioral leadership, ethical leadership, moral leadership, and managerial-ethical leadership [42]. CSR was born from ethical leadership by management within it [35], where managers push the welfare of others [43]. Employees regard these leaders as the ethical role models to set high-performance standards [44]. They increase competence and internalize the meaning and impact of their work by applying work ethics in the company [45,46]. Scholars have agreed that ethical leaders support corporate sustainability [47] by driving profitability [48]. These leaders practice what they preach, avoid injustice, and communicate meaningful information [49]. They are ethical behavior models [50,51] that explain employee honesty, integrity, and trust [52]. Employees can be rewarded for their ethical behavior [53] or punished for their unethical behavior in the workplace [54,55] by these leaders.

Transformational leadership frames change as an opportunity for employees' personal growth through the process of transitioning to a new environment [56]. Transformational leaders identify company goals and interests based on environmental needs [57] without maintaining the status quo. They influence employees to change in various ways [58] and encourage employees to adopt the company's vision [59] by paying special attention, understanding needs, and providing emotional support in the workplace [60]. Researchers have broadly discussed inspirational communication, intellectual stimulation, supportive leadership, personal recognition, and vision as transformational leadership characteristics [61–65].

Inclusive leadership supports team innovation [66] by promoting a sense of inclusivity, access, openness, and quality [67]. It encourages tolerance and acceptance [68,69] in the cooperation among diverse team members [70]. It is also emotional supportive for employees, increasing trust [46], openness, and access by leaders [71]. The leaders push employees to use alternatives to achieve the best results [72]. Inclusive leaders become role models for employees [73] who provide freedom and autonomy to perform responsibilities at work [74]. This inclusivity is crucial for the success of CSR.

The concept of responsible leadership developed from a study of corporate responsibility [75]. The leaders are responsible for reporting transparency on the company's economic, social, and ecological performance [76]. They concentrate on social goals towards stakeholders [77–82]. This leadership transmits relevant information and interacts with employees [83], provides an example for employees about doing things the right way in making decisions at work [84,85], and intensively responds to employee concerns [86].

Recent studies have broadly revealed leadership styles and HRMPs for employee performance in companies, for example, transformational leadership and job performance [57], transformational leadership and job performance [87], transformational leadership and sustainable employee performance [59], transformational leadership and employee performance [88], employee leadership and performance [89,90], leadership and work performance [91], transformational leadership and employee performance [92], spiritual leadership and employee performance [93], authoritarian leadership and employee performance [94], strategic leadership and employee performance [95], and others. However, the investigations in these studies have separated HRMPs in companies.

**Hypothesis (H1).** *SLSs of the CEO of the HRM consulting firm in the context of ethical leadership, transformational leadership, inclusive leadership, and responsible leadership positively affect CP.*

2.2.2. Collaborative HRM, High-Involvement HRM, Commitment-Based HRM, CP

Collaboration, involvement, and commitment of employees to work support the company's CSR programs. Researchers have revealed that collaborative HRM is about recruitment, training, performance appraisal, and rewards that facilitate the exchange of information, trust, and collaboration among employees [96]. The work is to develop cross-functional teams and close coordination between different organizational units within the company [97]. This collaboration is for economic, social, and environmental benefits, through team building, training, communication mechanisms, job rotation, mentoring relationships, and knowledge transfer.

High-involvement HRM relieves employees determine their destiny and encourages HR in developing societies. Researchers agree that high-involvement HRM concentrates on employee engagement through developing their knowledge, skills, and abilities directly on company problems [98–100]. Four HRM practices with high involvement as fundamental factors by companies are training, the autonomy of work methods, information, and participation in strategic decision making [101]. This system has enhanced employee skills, incentives, participation [102], and employee affective commitment [103].

Strategic HRM is a practice that commits to long-term investment in employees and a relational view of the working relationship [104,105]. A commitment-based HR system brings talented employees into advanced training programs [106,107], compensation programs, performance appraisals, and employee participation [108]. Scholars have agreed that HRM forges a psychological connection between employees and organizational goals [109].

**Hypothesis (H2).** *SHRMPs in the context of collaborative HRM, high-involvement HRM, and commitment-based HRM positively affect CP.*

*2.3. Leadership Styles, HRMPs, CJS, and CP*

Employee job satisfaction is a sequence of psychology [110]. The level of employee job satisfaction increases depending on their job [111,112]. Scholars have justified that job satisfaction is a psychological evaluation material [113]. Employee job satisfaction intrinsically refers to the performed job and the tasks performed by the employee [114]. Extrinsically, scholars have acknowledged that employee job satisfaction refers to the desired recognition, compensation, promotion, salary, rewards, work associations, and work components [115,116]. Researchers have admitted that employee job satisfaction concentrates on needs, motives, and goals [117]. It is associated with transformational leadership [118]. Several studies have also revealed the relationship between employee performance and transformational leadership [88], employee job satisfaction and ethical leadership [116], and employee job satisfaction and HR flexibility [119]. Employee job satisfaction positively effects employee performance [87,120–122].

**Hypothesis (H3).** *SLS of the CEO of the HRM consulting firm, in the context of ethical leadership, transformational leadership, inclusive leadership, and responsible leadership, positively affect CJS.*

**Hypothesis (H4).** *SHRMPs in the context of collaborative HRM, high-involvement HRM, and commitment-based HRM positively affect CJS.*

**Hypothesis (H5).** *CJS positively affects CP.*

**Hypothesis (H6).** *CJS mediates the relationship between the SLSs of the CEO of the HRM consulting firm in the context of ethical leadership, transformational leadership, inclusive leadership, responsible leadership, and CP.*

**Hypothesis (H7).** *CJS mediates the relationship between SHRMPs in the context of collaborative HRM, high-involvement HRM, commitment-based HRM, and CP.*

Figure 1 is the research model based on the hypotheses. SLSs and SHRMPs are two independent variables. CJS is the mediating variable, and CP is the dependent variable in the model. The four leadership styles are ethical leadership, transformational leadership, inclusive leadership, and responsible leadership. The three HRMPs are collaborative HRM, high-involvement HRM, and commitment-based HRM. The two types of CJS are intrinsic and extrinsic job satisfaction. The two types of CP are individual and team performance.

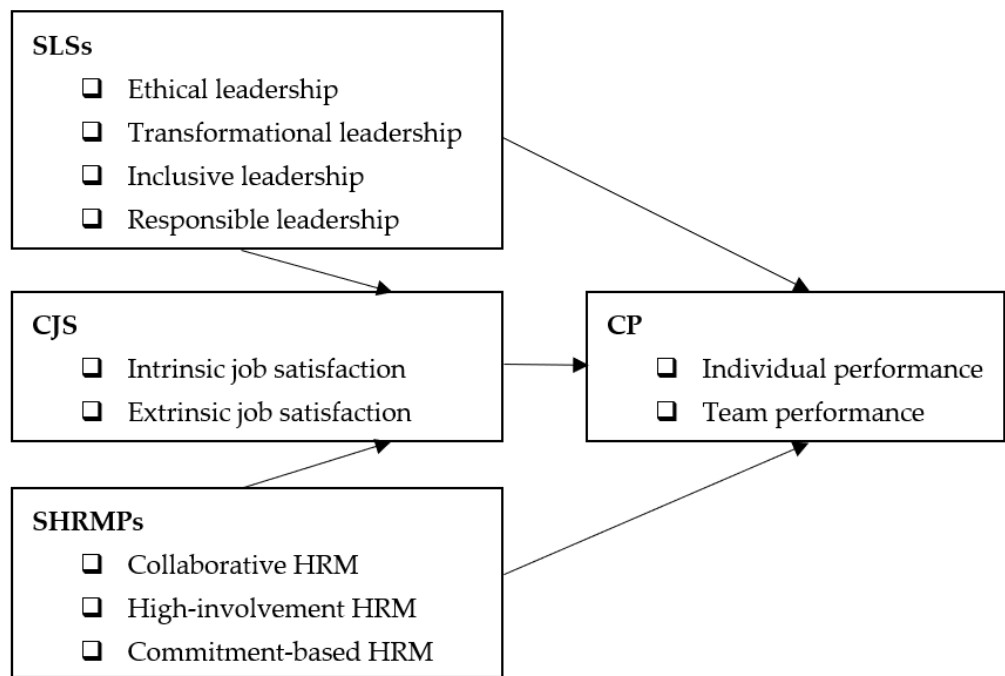

**Figure 1.** A causal relationship between SLSs, SHRMPs, CJS, and CP.

## 3. Data and Methods

### 3.1. Procedures and Respondent Profile

We distributed self-report questionnaires to HRM consultants working in different HRM consulting firms in Indonesia in early January 2022. We used the Snowball sampling method to achieve the target of four hundred consultants. We sent emails to the email addresses of these firms and asked them to voluntarily share the questionnaire links with the HRM consultants working with them. There was no limit to the number of consultants we set for each management consulting firm. It was not used as a one-respondent survey questionnaire.

The questionnaire was anonymous; we did not ask respondents' names nor the names of their company. It contained five questions for the respondent's profile, namely gender, age, formal education, work experience, and certification as in Table 1, and 27 statements

filled in by all respondents using 7 Likert scales ranging from "strongly disagree" to "strongly agree" as in Table 2.

**Table 1.** Profile of respondent.

| Demography | Percentage (%) | | | | |
|---|---|---|---|---|---|
| Gender | Female (23.8) | Male (76.2) | | | |
| Age (years old) | 20–30 (5.9) | 31–40 (9.1) | 41–50 (43.3) | 51–60 (25.9) | >60 (15.8) |
| Formal education (degrees) | Bachelor (9.1) | Master (76.7) | Doctoral (14.2) | | |
| Work experience (year) | <5 (15.0) | 5–10 (50.8) | 10–15 (16.6) | 15–20 (10.2) | >20 (7.5) |
| Certification | No (19.5) | Yes (80.5) | | | |

**Table 2.** Self-report questionnaire development.

| No | Items or Indicators | Concept and Sources |
|---|---|---|
| | *The CEOs of HRM consulting firms leads the consultants by . . . . . . . . .* | |
| 1 | Prioritizing integrity (X1.1). | |
| 2 | Building harmony (X1.2). | Ethical leadership [44,50,51]. |
| 3 | Being an ethical role model (X1.3). | |
| 4 | Being a change agent (X1.4). | Transformational leadership [56,61–65]. |
| 5 | Prioritizing intellect (X1.5). | |
| 6 | Supporting individual differences (X1.6). | |
| 7 | Providing good communication access (X1.7). | Inclusive leadership [67,70]. |
| 8 | Supporting diverse teamwork (X1.8). | |
| 9 | Maintaining long-term relationships with internal stakeholders (X1.9). | Responsible leadership [76–82]. |
| 10 | Maintaining long-term relationships with external stakeholders (X1.10). | |
| | *HR managers of HRM consulting firms . . . . . . . . .* | |
| 11 | Consider collaboration skills in recruiting consultants. (X2.1). | |
| 12 | Consider collaboration skills in assessing performance and compensating consultants (X2.2). | Collaborative HRM [98,99]. |
| 13 | Maintain a long-term relationship with external parties (X2.3). | |
| 14 | Put forward the sharing of information and feedback in the work of the consultant team (X2.4). | High-involvement HRM [100–103]. |
| 15 | Provide opportunities for internal parties in the recruitment and selection of consultants. (X2.5). | |
| 16 | Pay attention to the commitment to providing consultant compensation (X2.6). | Commitment-based HRM [108,109]. |
| 17 | Provide team collaboration training for consultants (X2.7). | |
| | *HRM consultants satisfied with . . . . . . . . .* | |
| 18 | Their works in the firms (Y1.1). | Intrinsic job satisfaction [116]. |
| 19 | The income their received from the firms.(Y1.2). | |
| 20 | Promotions their obtained from the firms (Y1.3). | Extrinsic job satisfaction [117,118]. |
| 21 | Supervision in their work in the firms (Y1.4). | |
| 22 | Colleagues in their work in the firms (Y1.5). | |
| | *The HRM consultants have . . . . . . . . .* | |

**Table 2.** *Cont.*

| No | Items or Indicators | Concept and Sources |
|----|--------------------|---------------------|
| 23 | High motivation to work in the firms (Y2.1). | Individual employee performance [30,31,33]. |
| 24 | High productivity at work in the firms (Y2.2). | |
| 25 | High task performance in working in the firms (Y2.3). | |
| 26 | Good teamwork knowledge at work (Y2.4). | Team employee performance [32]. |
| 27 | A great contribution to teamwork (Y2.5). | |

Note: " . . . " represents different influencing indicators.

The steps in data collection started from randomly collecting websites of HRM consulting firms in Indonesia through the Google search engine with the keyword "Business Management Consultants in Indonesia." After that, we sent emails to the email addresses of those companies. The subject was a request to distribute the questionnaire to consultants who work on it to access. The content was a link for them to access. Finally, we pooled 386 responses from consultants of different consultancies after four weeks, and 374 of them were complete responses.

Table 1 summarizes the key characteristics of our respondents. The sample data presented in the table are consistent with the aggregation of consultants working in the Indonesian consulting industry.

*3.2. Measurement*

Table 2 contains the 27 items in the questionnaire. First, we operate 10 indicators of SLSs adapted from concepts of ethical leadership, transformational leadership, inclusive leadership, and responsible leadership in current research to measure SLSs of 50 CEOs of HRM consulting firms in Indonesia. Indicators of ethical leadership are integrity, harmony, and ethical role models, while indicators of transformational leadership are change agent and intellectuality. Indicators of inclusive leadership are individual differences, communication access, and diverse teamwork, while items of responsible leadership are long-term relationships with internal and external stakeholders.

Then, seven indicators of SHRMPs were adapted from the concepts of collaborative HRM, high-involvement HRM, and commitment-based HRM in current research to measure HRMPs of the 50 firms. Collaborative HRM is indicated by collaboration skills in recruiting consultants, assessing performance, compensating consultants, and maintaining a long-term relationship with external parties. Indicators of high-involvement HRM are the sharing of information and feedback in the work of the consultant team and providing opportunities for internal parties in the recruitment and selection of consultants. Commitment-based HRM is measured by attention to the commitment to providing consultant compensation and provides team collaboration training for consultants.

Third, five indicators of CJS are adapted from concepts of intrinsic and extrinsic employee job satisfaction in current research to measure the job satisfaction of consultants who work in the 50 firms. Their work is an indicator for their intrinsic job satisfaction in the firms. Their extrinsic job satisfaction is indicated by the income they received from the company, promotions they got from the company, supervision in their work in the company, colleagues in their work in the company.

Finally, five CP indicators are from the concept of individual and team employees in current research to measure CP of those who work in the firms. Their performance in the context of the individual is indicated by their high motivation to work, productivity at work, and task performance in working in the firms. Indicators of their team performance are good teamwork knowledge and contribution to teamwork.

## 4. Results

### 4.1. Descriptive Analysis

Table 3 contains the mean, standard deviation, and variance of all indicators. Indicators of SLSs are X1.1 to X1.10. X2.1 to X2.7 are indicators of SHRMPs. Five indicators of CJS are Y1.1 to Y1.5. Y2.1 to Y2.5 are indicators of CP. A total of 374 respondents completely answered all items or indicators. Their answers ranged between strongly disagree (1) to strongly agree (7). The mean value of respondents' answers ranged from 4.58 to 5.01, with a standard deviation ranging from 1.515 to 1.725. The variance value ranged from 2.296 to 2.977, meaning that the average respondents have slightly agreed with all items in the questionnaire. Respondents' answers have spread from a scale of 1: strongly disagree, 2: disagree, 3: slightly disagree, 4: neutral, 5: slightly agree, 6: agree, to 7: strongly agree.

**Table 3.** Descriptive statistics.

| Items/Indicators | N | Minimum | Maximum | Mean | Standard Deviation | Variance |
|---|---|---|---|---|---|---|
| X1.1 | 374 | Strongly disagree (1) | Strongly Agree (7) | 4.64 | 1.628 | 2.650 |
| X1.2 | | | | 4.58 | 1.585 | 2.512 |
| X1.3 | | | | 4.74 | 1.725 | 2.977 |
| X1.4 | | | | 4.69 | 1.548 | 2.397 |
| X1.5 | | | | 4.70 | 1.622 | 2.631 |
| X1.6 | Idem | Idem | Idem | 4.77 | 1.622 | 2.629 |
| X1.7 | | | | 4.68 | 1.568 | 2.460 |
| X1.8 | | | | 4.85 | 1.615 | 2.608 |
| X1.9 | | | | 4.76 | 1.612 | 2.597 |
| X1.10 | | | | 4.87 | 1.611 | 2.595 |
| X2.1 | | | | 4.70 | 1.618 | 2.618 |
| X2.2 | | | | 4.81 | 1.661 | 2.760 |
| X2.3 | | | | 4.80 | 1.640 | 2.690 |
| X2.4 | Idem | Idem | Idem | 4.80 | 1.624 | 2.638 |
| X2.5 | | | | 4.78 | 1.614 | 2.604 |
| X2.6 | | | | 4.80 | 1.581 | 2.499 |
| X2.7 | | | | 4.84 | 1.645 | 2.707 |
| Y1.1 | | | | 5.01 | 1.613 | 2.600 |
| Y1.2 | | | | 4.85 | 1.610 | 2.592 |
| Y1.3 | Idem | Idem | Idem | 4.77 | 1.554 | 2.413 |
| Y1.4 | | | | 4.77 | 1.515 | 2.296 |
| Y1.5 | | | | 4.68 | 1.516 | 2.297 |
| Y2.1 | | | | 4.83 | 1.548 | 2.398 |
| Y2.2 | | | | 4.75 | 1.574 | 2.478 |
| Y2.3 | Idem | Idem | Idem | 4.82 | 1.530 | 2.340 |
| Y2.4 | | | | 4.78 | 1.524 | 2.322 |
| Y2.5 | | | | 4.78 | 1.533 | 2.352 |

### 4.2. Confirmatory Factor Analysis

This quantitative research uses an explanatory approach with a covariance-based structural equation model (CB-SEM). This model combines factor analysis and path analysis

from theory. Therefore, for accurate analysis results and a good model image, we used AMOS 23. Before analyzing the model, confirmatory factor analysis to confirm the feasibility of the measurements was performed.

For indicators' validity, in Table 4, the construct validity test shows that the critical ratio value of all items is higher than 1.96, and the probability value is less than 0.05. This means that all items are valid. In Table 5, the convergent validity test shows that all loading factor values are from 0.799 to 0.901 (>0.5), and the validity test using average variance extract (AVE) with the standard loading$^2$/$\sum$standard loading$^2$ + $\sum$ej formula shows that all values are above 0.5. It also indicates that all items are valid. The discriminant validity test in Table 6 shows that the values based on the implied covariance metric are higher than the correlation values between the surrounding variables (on the left and below). So, all items used are valid. For item reliability, the reliability test with Cronbach Alpha (CA) and construct reliability test (CR) with the formula ($\sum$standard loading)$^2$/($\sum$standard loading)$^2$ + $\sum$ej illustrate that all values are above 0.7; this means all items are reliable.

**Table 4.** Construct validity test.

| Items/Indicators | Critical Ratio | Probability | Items/Indicators (Continued) | Critical Ratio (Continued) | Probability (Continued) |
|---|---|---|---|---|---|
| X1.1 ← SLSs | | | Y1.1 ← CJS | | |
| X1.2 ← SLSs | 22.935 | *** | Y1.2 ← CJS | 22.932 | *** |
| X1.3 ← SLSs | 21.462 | *** | Y1.3 ← CJS | 21.822 | *** |
| X1.4 ← SLSs | 22.942 | *** | Y1.4 ← CJS | 23.501 | *** |
| X1.5 ← SLSs | 20.610 | *** | Y1.5 ← CJS | 21.286 | *** |
| X1.6 ← SLSs | 22.823 | *** | Y2.1 ← CP | | |
| X1.7 ← SLSs | 19.480 | *** | Y2.2 ← CP | 20.881 | *** |
| X1.8 ← SLSs | 20.790 | *** | Y2.3 ← CP | 20.333 | *** |
| X1.9 ← SLSs | 20.814 | *** | Y2.4 ← CP | 20.447 | *** |
| X1.10 ← SLSs | 23.430 | *** | Y2.5 ← CP | 18.069 | *** |
| X2.1 ← SHMRPs | | | | | |
| X2.2 ← SHRMPs | 23.451 | *** | | | |
| X2.3 ← SHRMPs | 22.016 | *** | | | |
| X2.4 ← SHRMPs | 23.001 | *** | | | |
| X2.5 ← SHRMPs | 22.759 | *** | | | |
| X2.6 ← SHRMPs | 22.407 | *** | | | |
| X2.7 ← SHRMPs | 20.738 | *** | | | |

*** = probability < 0.001.

### 4.3. Goodness of Fit and Structural Equation Modeling

The results of the model fit test ensure that the research model is fit to proceed to the hypothesis testing process with SEM. Model Chi Square χ2 or CMIN = 525.477, degree of freedom = 318, CMIN/df = 1.652 (<2), root mean square error of approximation (RMSEA) = 0.42 (<3), comparative fit index (CFI) = 0.980 (>0.9), Truker–Lewis fit index (TLI) 0.978 (>0.9), incremental fit index (IFI) 0.980 (>0.9), relative fit index (RFI) = 0.945 (>0.9), normal fit index NFI = 0.950 (>0.9), Root Mean Square Residua (RMR) = 0.57 (>0.9), Goodness of Fit (GFI) = 0.908 (>0.9), and PRATIO = 0.906 (>0.9). Figure 2 describes the SEM output of AMOS. The model is recursive with a sample size of 374. The number of distinct sample moments is 378. The number of variables is 60: 27 observed variables and 33 unobserved variables. It can also be 31 exogenous variables and 29 endogenous variables.

**Table 5.** Convergent validity, construct reliability, average variance extracted.

| Items/ Indicators | Loading Factors | Construct Reliability (CR) and Cronbach's Alpha (CA) | Average Variance Extracted (AVE) | Indicators (Continued) | Loading Factors (Continued) | Construct Reliability (CR) and Cronbach's Alpha (CA) (Continued) | Average Variance Extracted (AVE) (Continued) |
|---|---|---|---|---|---|---|---|
| X1.1 | 0.842 | | | Y1.1 | 0.901 | | |
| X1.2 | 0.887 | | | Y1.2 | 0.839 | | |
| X1.3 | 0.855 | | | Y1.3 | 0.817 | 0.836 (CR), 0.925 (CA) | 0.504 |
| X1.4 | 0.886 | | | Y1.4 | 0.848 | | |
| X1.5 | 0.835 | 0.914 (CR), 0.965 (CA) | 0.517 | Y1.5 | 0.810 | | |
| X1.6 | 0.884 | | | Y2.1 | 0.824 | | |
| X1.7 | 0.807 | | | Y2.2 | 0.873 | | |
| X1.8 | 0.838 | | | Y2.3 | 0.855 | 0.839 (CR), 0.924 (CA) | 0.510 |
| X1.9 | 0.840 | | | Y2.4 | 0.864 | | |
| X1.10 | 0.897 | | | Y2.5 | 0.799 | | |
| X2.1 | 0.858 | | | | | | |
| X2.2 | 0.881 | | | | | | |
| X2.3 | 0.855 | | | | | | |
| X2.4 | 0.876 | 0.884 (CR), 0.953 (CA) | 0.522 | | | | |
| X2.5 | 0.870 | | | | | | |
| X2.6 | 0.861 | | | | | | |
| X2.7 | 0.829 | | | | | | |

**Table 6.** Metric implied covariance.

| | SHRMPs | SLSs | CJS | CP | | | | | | |
|---|---|---|---|---|---|---|---|---|---|---|
| SHRMPs | 1.920 | | | | | | | | | |
| SLSs | 1.631 | 1.873 | | | | | | | | |
| CJS | 1.482 | 1.352 | 2.104 | | | | | | | |
| CP | 1.066 | 1.014 | 1.694 | 1.623 | | | | | | |
| | X1.10 | X1.9 | X1.8 | X1.7 | X1.6 | X1.5 | X1.4 | X1.3 | X1.2 | X1.1 |
| X1.10 | 2.588 | | | | | | | | | |
| X1.9 | 1.952 | 2.591 | | | | | | | | |
| X1.8 | 1.952 | 1.829 | 2.601 | | | | | | | |
| X1.7 | 1.825 | 1.710 | 1.710 | 2.453 | | | | | | |
| X1.6 | 2.067 | 1.936 | 1.936 | 1.810 | 2.622 | | | | | |
| X1.5 | 1.953 | 1.830 | 1.830 | 1.711 | 1.937 | 2.624 | | | | |
| X1.4 | 1.978 | 1.853 | 1.853 | 1.732 | 1.962 | 1.854 | 2.390 | | | |
| X1.3 | 2.127 | 1.993 | 1.993 | 1.863 | 2.110 | 1.994 | 2.020 | 2.969 | | |
| X1.2 | 2.027 | 1.899 | 1.899 | 1.775 | 2.010 | 1.900 | 1.924 | 2.069 | 2.505 | |
| X1.1 | 1.975 | 1.851 | 1.851 | 1.730 | 1.959 | 1.852 | 1.875 | 2.017 | 1.921 | 2.643 |
| | X2.7 | X2.6 | X2.5 | X2.4 | X2.3 | X2.2 | X2.1 | | | |
| X2.7 | 2.7 | | | | | | | | | |

**Table 6.** *Cont.*

|  | SHRMPs | SLSs | CJS | CP |  |  |  |
|---|---|---|---|---|---|---|---|
| X2.6 | 1.85 | 2.492 |  |  |  |  |  |
| X2.5 | 1.909 | 1.906 | 2.597 |  |  |  |  |
| X2.4 | 1.935 | 1.931 | 1.992 | 2.631 |  |  |  |
| X2.3 | 1.907 | 1.903 | 1.964 | 1.99 | 2.683 |  |  |
| X2.2 | 1.991 | 1.987 | 2.051 | 2.078 | 2.048 | 2.753 |  |
| X2.1 | 1.887 | 1.883 | 1.943 | 1.969 | 1.941 | 2.026 | 2.611 |
|  | Y1.5 | Y1.4 | Y1.3 | Y1.2 | Y1.1 |  |  |
| Y1.5 | 2.291 |  |  |  |  |  |  |
| Y1.4 | 1.573 | 2.289 |  |  |  |  |  |
| Y1.3 | 1.553 | 1.626 | 2.407 |  |  |  |  |
| Y1.2 | 1.654 | 1.731 | 1.709 | 2.585 |  |  |  |
| Y1.1 | 1.778 | 1.862 | 1.838 | 1.957 | 2.593 |  |  |
|  | Y2.5 | Y2.4 | Y2.3 | Y2.2 | Y2.1 |  |  |
| Y2.5 | 2.345 |  |  |  |  |  |  |
| Y2.4 | 1.608 | 2.316 |  |  |  |  |  |
| Y2.3 | 1.598 | 1.719 | 2.334 |  |  |  |  |
| Y2.2 | 1.678 | 1.805 | 1.793 | 2.472 |  |  |  |
| Y2.1 | 1.558 | 1.675 | 1.665 | 1.748 | 2.391 |  |  |

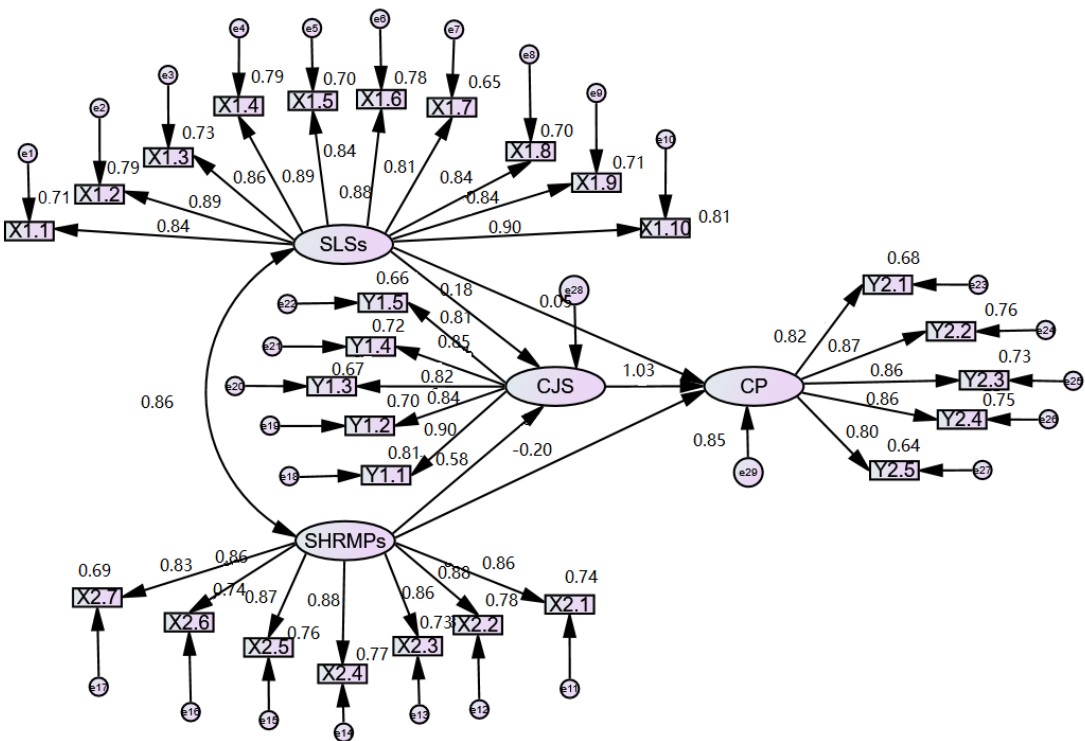

**Figure 2.** SEM data processing output with AMOS.

### 4.4. Hypotheses

Table 7 reports the results of the hypothesis testing based on the critical ratio and probability. Four of the seven hypotheses tested have been accepted while three hypotheses

have been rejected. On the one hand, the results of hypothesis testing using structural equation modeling show the acceptance of hypotheses 3, 4, 5, and 7, since their critical ratios are higher than 1.96 and their probability values are below 0.05. On the other hand, since hypotheses 1, 2, and 6 have critical ratios less than 1.96 and their probability values are above 0.05, the results reject them.

**Table 7.** Hypothesis testing.

| Direct Effect | Indirect Effect | Total Effect | Critical Ratio | Probability | Decision |
|---|---|---|---|---|---|
| H1: SLSs → CP = 0.053 | H6: SLSs → CJS → CP = 0.185 | 0.238 | 0.832 | 0.405 | H1 and H6 are rejected. |
| H2: SHRMPs → CP = −0.199 | H7: SHRMPs → CJS → CP = 0.598 | 0.399 | −2.745 | 0.006 | H2 is rejected but H7 is accepted. |
| H3: SLSs → CJS = 0.180 | | 0.180 | 2.126 | 0.034 | H3 is accepted. |
| H4: SHRMPs → CJS = 0.583 | | 0.583 | 6.641 | *** | H4 is accepted. |
| H5: CJS → CP = 1.027 | | 1.027 | 16.187 | *** | H5 is accepted. |

*** = probability < 0.001.

SLSs positively affect CJS with a critical ratio of 2.126 and a probability value of 0.034 (<0.05). SHRMPs positively affect CJS with a critical value of 6.641 and a probability value of 0.000 (<0.05). CJS positively affects CP with a critical value of 16,187 and a probability value of 0.000 (<0.05). Thus, CJS is a mediator that increases the positive effect of SHRMPs on CP.

Hypothesis 5 has the highest direct and total effect. It has a positive effect of CJS on CP with a value of 1.027. Hypothesis 7 has the highest indirect effect, namely the positive effect of SHRMPs on CP mediated by CJS, which is 0.598. Full mediation occurs in an indirect relationship between SHRMPs, CJS, and CP (Hypothesis 7), wherein it is the case in which SHRMP practices no longer positively affect CP after CJS creates a pathway.

## 5. Discussion

Based on Table 5, the indicators with the highest loading factor values are X1.10, X2.2, Y1.1, and Y2.2. They are the responsible leadership in maintaining long-term relationships with external stakeholders, consider collaboration skills in assessing performance and compensating consultants, the consultants' satisfaction with their work, and high productivity at work, respectively.

The high motivation, productivity, and task performance of HRM consultants in Indonesia reflects their high performance individually. Their good teamwork knowledge and great team contributions reflect their performance in teamwork. The consultants' high performance is positively affected by their CEO's SLSs and the CJS. Their job satisfaction is positively affected by their CEO's SLS and SHRMPs. The CP can be positively affected by SHRMPs if the CJS mediates the relationship.

Although Indonesia does not yet have a national consulting index (NCI) among other Asian countries, such as Bangladesh, China, Taiwan, Hong Kong, India, Japan, South Korea, the Philippines, Singapore, and Thailand [23], the management consulting industry in Indonesia is dynamic with good consultant performance even amid the COVID-19 pandemic. The leadership styles and HRM practices in the firms in Indonesia have been sustainable. The CEOs of these firms have applied ethics, transformation, inclusion, and responsibility in their leadership. The HR managers of these firms have adopted collaboration, high involvement, and commitment in HRMPs in the firms. These SLSs and SHRMPs determine job satisfaction and the performance of the consultants who work in the firms.

Ethics, transformation, inclusion, and responsibility have been pillars of sustainable leadership for HRM consulting firms in Indonesia. Collaboration, high involvement, and commitment have been the foundations for HRM practices for HRM consulting firms in

Indonesia. Their SLSs and SHRMPs have become crucial aspects for the job satisfaction and performance of the consultants who work in them.

### 5.1. Theoretical Implications

This research model is consistent with the theories that explain that collaborative, high involvement, and commitment-based HRM positively affect employee job satisfaction and employees. It is also steady with premises that put forward leadership styles that positively affect employee job satisfaction, later employee job satisfaction, employee performance. However, it is antithetical to the theories that support ethical, transformational, inclusive, and responsible leadership that positively affects employee performance. This research model is inconsistent with the positive effect of leadership styles and HRMPs on employee performance.

In the context of leadership, this study comes around with the study by Leroy et al. (2018) that explained the interactions between leadership styles and HRMPs for employee motivation and productivity. The results of this study are in line with the concepts of the interaction between leadership and HR behavior and practices for employee task performance [31], responsible leadership that interacts with HRM [33], researchers have revealed that ethical leadership is related to CSR [35,36], transformational leadership related to CSR [37]. inclusive leadership related to CSR [38].

In the context of HRM, this study supports studies that explain that researchers have agreed that collaborative HRM is the link for companies [39], and individual commitment to the environment must begin with employee selection based on mental criteria for the environment and employee involvement in environmental management that supports the company's environmental performance. Collaborative HRM takes into account economic, social, and environmental balance through team building, training, communication mechanisms, job rotation, mentoring relationships, and knowledge transfer [40]. The results of this study are in line with the explanations that researchers have agreed that high-involvement HRM concentrates on employee engagement through developing their knowledge, skills, and abilities directly on company problems [98–100], strategic HRM is characterized by HRMPs committed to long-term investment in employees and a relational view of the working relationship [104,105], a commitment-based HR system brings talented employees into advanced training programs [106,107], compensation programs, performance appraisal, and employee participation [108]; researchers have agreed that HRM establishes a psychological relationship between employees and organizational goals [109].

In the context of employee job satisfaction, this research is in line with explanations about researchers that have admitted employee job satisfaction concentrates on needs, motives, and goals [117]; employee job satisfaction is related to their performance [118]. In the context of job satisfaction and employee performance, this study is consistent with explanations about several studies that have revealed the relationship between employee job satisfaction and transformational leadership [87], the relationship between employee job satisfaction and ethical leadership [120], employee job satisfaction and HR practices [60,119], and employee job satisfaction and employee performance [121,122].

For further research, "SLSs and SHRMPs positively affect employee job satisfaction" will be a hypothesis. The concept of ethical, transformational, inclusive, and responsible leadership can be tested separately or compared in positively affected employee job satisfaction. Collaborative, high involvement, and commitment-based HRMPs that positively affect employee job satisfaction and performance can be compared. The model of these relationships can be tested into other objects such as hospitals, hotels. Although job satisfaction is not a mediator in the relationship between SLSs and CP, this in-direct relationship can be tested again with different objects. A comparison between two or more different object locations can be done to determine whether the results of hypothesis testing will remain robust. Aspects of national culture can determine the level of leadership styles and HRMPs.

*5.2. Practical Implications*

First, it is a demand for the CEOs of HRM consulting firms to apply ethical, transformational, inclusive, and responsible aspects to accentuate sustainability in their leadership styles. They need to enlist harmonization between profit, environment, and society. It can increase the performance of consultants who are the leading employees. Second, it is a competitive advantage for HR managers of HRM consulting firms to capture collaboration, high involvement, and commitment for SHRMPs. This practice can increase their CJS and further improve their CP. Third, the job design of consultants is to consider maintaining their job satisfaction.

The sustainability in leadership styles and HRMPs can be the added values for the struggle of the HRM consulting business competition in Indonesia, where the industry is growing even amid the COVID-19 pandemic. They can present features about the sustainability of their leadership styles and HRM practices by marketing their services through official websites and social media networks.

The consultants had a high performance, both individually and as a team. First, the CEOs of the firms need to prioritize responsible leadership by maintaining long-term relationships with external stakeholders. Second, HR managers need to prioritize commitment-based HRM by providing team collaboration training to consultants. Third, consultants need to prioritize their intrinsic performance by increasing job satisfaction with their work. SLSs, including ethical leadership, transformational leadership, inclusive leadership, and responsible leadership, have been applied to improve consultant performance in the firms. SHRMPs, including collaborative HRM, high-involvement HRM, and commitment-based HRM, have been implemented to increase job satisfaction performance of the consultants.

The rise of HRM consulting companies in Indonesia can be balanced with the continuity of their leadership and HRM. When the CP is high, this indicates the high level of sustainability in leadership in this consulting firm can be seen from the high level of ethics, transformation, inclusion, responsibility in leadership, and the high sustainability of HRM in these firms can be seen from the high collaboration, involvement, and commitment of employees.

## 6. Conclusions

The leaders and HR managers of HRM consulting firms in Indonesia can promote sustainability in their leadership and HRM through ethics, transformation, inclusivity, responsibility, collaboration, high involvement, and commitment. These leadership styles and HRMPs positively affect the job satisfaction of consultants. However, they do not positively affect the consultant's performance. CJS does not mediate the relationship between the CEOs' leadership styles and CP. However, it mediates the relationship between SHRMPs and CP. In pursuing CP, SHRMPs are independent variables. Job satisfaction acts as a mediating one.

The results of this study can develop concepts and research on sustainability where, currently, sustainability can no longer be avoided by most companies. Global consulting firms already have annual sustainability reports, a competitive advantage for the firms. Since it is still rare to find similar research results, the results of this study are a good reference for developing the concept of ethical, transformational, inclusive, and responsible leadership in the context of sustainability in Indonesia.

The results of this study are valuable for concept development and research on collaborative, high-involvement, and commitment-based HRM, job satisfaction, and employee performance. They are valuable for the management of consulting firms in Indonesia in realizing the sustainability of their business, especially in preparing themselves to enter the post-COVID-19 zone by maintaining the aspects of ethics, transformation, inclusion, and responsibility. They become a guideline for them to continue to be sustainable by maintaining collaborative practices, high involvement, and commitment in HRM.

*Limitations*

Conceptually, this research only uses job satisfaction as a mediating variable. Many researchers in the field of the organization have described the relationship between leadership and organizational commitment and leadership and employee well-being. Various concepts have measured employee job satisfaction.

Methodologically, this research focuses on the HRM consulting firm as the object. Management consulting firms in other fields, such as financial and strategic management consulting firms, have also developed rapidly in Indonesia and other developing countries. Data were only from Indonesia. A collection period of more than a month would be preferable to collect more primary data from not a single country. More data will produce better results. The management consulting industry is well-developed in many other developing countries. Finally, this study cannot prove that SLSs and HRMPs positively affect CP and the mediation of job satisfaction between SLSs and CP.

**Author Contributions:** Conceptualization, A.C., J.P. and K.S.; methodology, A.C.; software, A.C.; validation, A.C., J.P. and K.S.; formal analysis, A.C.; investigation, A.C.; resources, A.C.; data curation, A.C.; writing—original draft preparation, A.C.; writing—review and editing, A.C. and J.P.; visualization, A.C.; supervision, J.P. and K.S. All authors have read and agreed to the published version of the manuscript.

**Funding:** This research received no external funding.

**Institutional Review Board Statement:** Ethical review and approval were waived for this study due to the fact that we used anonymous data that were not traceable to individuals at any time.

**Informed Consent Statement:** It is waived for this study due to the fact that we used anonymous data that were not traceable to individuals at any time.

**Conflicts of Interest:** We declare no conflict of interest.

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
