# Peer review of "Pursuing Consultant Performance: The Roles of Sustainable Leadership Styles, Sustainable Human Resource Management Practices, and Consultant Job Satisfaction"

_sustainability, doi:10.3390/su14073967_

Round 1
Reviewer 1 Report
A very well written comprehensive piece. It presents relevant, coherent and well-founded sections. However, I suggest you edit (or have it edited by a professional editor) to fix some grammatical errors. I also recommend that the Discussions section be more elaborate, refer to previous theoretical and empirical studies and the Conclusions section should also highlight the work's contribution to the development of knowledge in the field.
Reviewer 2 Report
The title of this paper has already explained that what this paper discusses is actually the relationship between the four variables. The discussion of this relationship is innovative to a certain extent and has practical significance for the practice of human and natural management.However, this paper still has the following problems, which need to be further improved: 1. The object of this study is Indonesia's enterprises, then, it is necessary to further explain the external validity of this study, that is, whether the experience of Indonesia's enterprises is generalizable, and whether it is also suitable for other developing countries. This needs to be explained in the introduction section. 2. The citation of the full text needs to be revised. The current citation format results in incomplete sentences. E.g: [27] suggest the interactions between leadership styles and HRM practices in employee motivation and productivity. [27] Apparently a quote, however, the sentence is missing the subject. There are many such situations in the full text, which need to be revised. 3. In terms of research methods, this paper adopts an email survey, so it is necessary to explain that it is a real-name survey but an anonymous survey. If the survey is anonymous, how can it be determined that the person participating in the survey is the target sample? 4. The language still needs to be polished by professional institutions. For example, 298 sentences have a language disorder such as "Our Our team". 5.5. In terms of methods, structural equation modeling is used in this paper, but partial least squares structural equation modeling (PLS-SEM) is not used. Therefore, it is necessary to compare the similarities and differences of different methods and explain the rationality of the current method used in this paper. 6. The discussion part needs to be further expanded and strengthened, and more in-depth discussions need to be carried out in combination with relevant literature.Author Response
February 28, 2022.
To: Reviewer 2.
Dear reviewer,
Thank you for your assessment, comments, and suggestions on our article entitled Pursuing consultant performance: The role of sustainable leadership styles, sustainable human resource management practices, consultant job satisfaction. We have added a case study description for Indonesia in the Introduction section. We have added subjects to all citations that are missing. We have added an explanation of how to get respondents with anonymous questionnaires. Grammatical errors in our articles have been revised using Grammarly software and assisted by several English language experts. We have also explained the reasons and advantages of using CB-SEM with AMOS over PLS-SEM. We have included an elaboration in the Discussion section based on the theory and empirical studies used in the article.
All the improvements we made to your comments and suggestions became provision for our development in writing scientific articles.
Thank you for your attention and cooperation.
Kind regards,
Corresponding author.
Reviewer 3 Report
Dear authors, thank you for the effort you put into this paper. But, there are a few things I'd like to remark.First of all, you do need proofreading. Especially the first part of the paper contains too many mistakes in vocabulary. No doubt it's a matter of choosing the wrong translation, but still, please have your text checked.
Second, I know that the paper uses a numbering system for references, but the text would benefit from naming authors anyway. I had to go to the list of references multiple times to see what research you were referring to, showing the authors does help.
Thirdly, the conceptual model is much more clear in fig 3 than in fig 1, as it more clearly shows job satisfaction to be an intermediate. I would advice to revise fig 1.
As to abbrevations, these cannot always be avoided, but sometimes the abbreviations are a bit hard to interpret, see e.g. y.o. (table 1), and CR (Cronbach's alfa? Construct reliability? Critical ratio?)
I always put quite some effort in getting insight in the questionnaire for data collection. I was rather disappointed that you did not mention in paragraph 3.2 what sources you used to develop the questionnaire. Fortunately, you do show the questions in table 4, but I still have questions. First of all, you talk about different kinds of leadership, but I do not see any differentiation in leadership styes in your questionnaire. Do these questions reflect all leadership styles that you mentioned, and no other styles?? Furthermore, I wondered to what extent the respondents rated themselves, both for SHRMP, and CP. And I do hope that they answered questions on CJS for themselves. Could you provide more details on the survey questions? Were they phrased as presented in Table 4? And how to determine the validity of the data when you do not show the source of the questions?
I also wondered whether a simple Cronbach's alfa would not have been sufficient, provided that you use validated scales (I do not know whether the scales are validated, as you did not desribe where the sales are coming from). And the factor loadings look very nice, but you do not say whether there were any cross loading questions.
I also wondered why you did a regression (table 3) with the separate questions, not the constructs?
Table 2: Min, max and N are the same for all of the questions, so please put them in the heading. And with Likert scales you can help the reader by noting whether a high number is positive or negative (e.g. low or high job satisfaction).
Regarding the results: please check fig 3, it's hard to see what number corresponds with what line. And regarding table 6, given the probability of 0.034, why do you accept H3? Or did I misunderstand the table?
If I understand the results correctly, you found a full mediation by job satisfaction, but you did not call it FULL mediation. Though that's a nice result.
And you asked for several individual characteristics, but you did not use any of them in the analysis? And how representative is your sample? Could that habve influenced your results?
However, your theoretical implications are not theoretical implications, as you do not refer to a single source. You really do need to refer to relevant sources in this paragraph
All in all, I'd consider this a draft, not a fully developed paper yet.
Author Response
February 28, 2022.
To: Reviewer 3.
Dear reviewer,
Thank you for your assessment, comments, and suggestions on our article entitled Pursuing consultant performance: The roles of sustainable leadership styles, sustainable human resource management practices, consultant job satisfaction. We have corrected all grammatical and vocabulary errors in writing our article by using the Grammarly software and asking for help from several experts in writing in English. We have added subjects for all citations that are missing. We have changed the conceptual model to match the output of the AMOS model. We have fixed all the obscure abbreviations in the table. We have added a table of the contents of the questionnaire and a detailed description of the leadership styles and HRM practices. We have explained validity and reliability tests results. We have added the results of the Cronbach Alpha test. We have provided the indicators with the highest loading factors values. We have fixed all the obscure column headings, such as regression. We have also added an explanation for each table's contents. All errors in explaining our hypotheses are corrected. We have added a full mediation explanation. We have presented individual characteristic results. We have developed the discussion, the implications, and the conclusion.
All the improvements we have made to your comments and suggestions became provision for our development in writing scientific articles.
Thank you for your attention and contributions.
Kind regards,
Corresponding author.
Round 2
Reviewer 2 Report
The paper meets the standards.
Author Response
Dear reviewer,
Thank you for your assessment.
Point 1. We have revised the introduction and literature review sections. The relationship between content, theoretical background, and empirical research has become clearer.
Point 2. We have revised the results section. The arguments and discussions in finding have become coherent, balanced, and compelling.
Point 3. We have revised the conclusion section. The results and secondary references have supported it.
Thank you for your attention and understanding.
Kind regards,
Corresponding author
Reviewer 3 Report
Dear authors,
I'm happy to see that you revissedthe manuscript and took myremarks seriously. However, the Enhlish really needs to be improved. The quality of the language still prevents the reader from fully understanding your paper and enjoying reading it. The main problem is vocabulary.
An example: The purview of sustainable leadership flaps the accordof long-term vision, etc. Purview? Flaps? Accord?
Author Response
Dear reviewer,
Thank you for your assessment and comments.
Point 1. We have revised the introduction and literature review sections. The relationship between content, theoretical background, and empirical research has become clearer.
Point 2. We have revised the design, questions, hypotheses, and methods sections. They have become clearer.
Point 3. We have revised the results section. The arguments and discussions in finding have become clearer, balanced, and compelling.
Point 4. We have revised the results section to make it empirical research.
Point 5. We have revised the reference section. It has become clearer and more adequate.
Point 6. We have revised the conclusion section. It has been more supported by results and secondary references.
Thank you very much for your attention and understanding.
Kind regards,
Corresponding author
Round 3
Reviewer 3 Report
Dear authors, thank you for improving the language of the paper. For a final version you would stillneed to pay attention to the referencing, as you sometimes use [author, year] and sometimes [number] in the text